# DISTRIBUTIONAL REINFORCEMENT LEARNING WITH MONOTONIC SPLINES

**Yudong Luo**[1,4]**, Guiliang Liu**[1,4] **, Haonan Duan**[2,4]**, Oliver Schulte**[3]**, Pascal Poupart**[1,4]
[1]University of Waterloo, [2]University of Toronto, [3]Simon Fraser University, [4]Vector Institute
{yudong.luo,guiliang.liu,ppoupart}@uwaterloo.ca
haonand@cs.toronto.edu, oschulte@cs.sfu.ca

## ABSTRACT

Distributional Reinforcement Learning (RL) differs from traditional RL by estimating the distribution over returns to capture the intrinsic uncertainty of MDPs. One key challenge in distributional RL lies in how to parameterize the quantile function when minimizing the Wasserstein metric of temporal differences. Existing algorithms use step functions or piecewise linear functions. In this paper, we propose to learn smooth continuous quantile functions represented by monotonic rational-quadratic splines, which also naturally solve the quantile crossing problem. Experiments in stochastic environments show that a dense estimation for quantile functions enhances distributional RL in terms of faster empirical convergence and higher rewards in most cases.

## 1 INTRODUCTION

A fundamental problem in traditional value-based RL is to estimate the expectation of future returns (Mnih et al., 2015; Van Hasselt et al., 2016). Distributional RL differs from this by also taking into account the intrinsic randomness of returns within MDPs (Morimura et al., 2010; Bellemare et al., 2017). To do so, distributional RL algorithms characterize the total return as a random variable and estimate its underlying distribution. In contrast, traditional value-based RL algorithms focus only on the mean of the random variable.

Distributional RL offers several advantages over value-based RL that computes only expected returns. The distribution of returns enables risk-sensitive RL by facilitating the optimization of other statistics than just the mean of the returns (Dabney et al., 2018a; Martin et al., 2020). Even when we stick to maximizing the mean of the returns, the distribution offers a more reliable and robust way of computing the expectation, which has led to a series of records on the Atari benchmark among value-based non-distributed RL techniques (Bellemare et al., 2017; Dabney et al., 2018a;b; Hessel et al., 2018; Yang et al., 2019; Zhou et al., 2020; Nguyen et al., 2021). Intuitively, while it is sufficient to represent an expected return by a single mean value, errors due to finite samples and function approximations can be reduced by "canceling" each other when multiple sample returns or quantile values are used. This is similar to the benefits of ensemble learning techniques although, strictly speaking, distributional RL is not an ensemble RL technique. In fact, distributional RL has been combined with ensemble learning and truncated critic predictions to mitigate overestimation bias in continuous control (Kuznetsov et al., 2020).

One key aspect of distributional RL algorithms is the parameterization of return distributions. In Categorical DQN (C51) (Bellemare et al., 2017), the return distributions are limited to categorical distributions over a fixed set of discrete values. It is also shown that the distributional Bellman operator is a contraction under the maximal form of the Wasserstein metric, but in practice, C51 optimizes the cross-entropy loss with a Cramér-minimizing projection (Rowland et al., 2018). To bridge the gap between theoretical analysis and algorithmic implementation, quantile regression (QR)-based distributional RL algorithms (Dabney et al., 2018a;b; Yang et al., 2019; Zhou et al., 2020) estimate a finite number of quantile values instead of the distribution of returns since quantile regression can easily use the Wasserstein metric as the objective. In fact, the Wasserstein metric is approximately minimized by optimizing the quantile Huber loss (Huber, 1992) between the Bellman updated distribution and the current return distribution.

Although with an infinite number of quantiles, the step quantile function in those quantile regression based methods will approximate the full quantile function arbitrarily closely, in practice, it is infeasible to have infinite quantiles in most existing architectures. In addition, the quantile crossing issue, recently pointed out and solved by (Zhou et al., 2020), was ignored by previous distributional RL techniques. The issue is that if no global constraint is applied, the quantile values estimated by a neural network at different quantile levels are not guaranteed to satisfy monotonicity, which can distort policy search and affect exploration during training (Zhou et al., 2020).

In this work, we propose to learn a continuous representation for quantile functions based on monotonic rational-quadratic splines (Gregory & Delbourgo, 1982). The monotonic property of these splines naturally solves the quantile crossing issue described above. Furthermore, unlike step functions or piecewise linear functions that provide a crude approximation in each bin, monotonic splines provide a more flexible and smooth approximation. With sufficiently many knots, splines can approximate any quantile function arbitrarily closely. We compare empirically our spline-based technique with other quantile-based methods in stochastic environments. We demonstrate that our method offers greater accuracy in terms of quantile approximation, faster convergence during training and higher rewards at test time.

## 2 DISTRIBUTIONAL REINFORCEMENT LEARNING

In standard RL settings, agent-environment interactions are modeled as a MDP, represented as a tuple $(\mathcal{S}, \mathcal{A}, R, P, \gamma)$ (Puterman, 2014). $\mathcal{S}$ and $\mathcal{A}$ denote state and action spaces. $P(\cdot|s, a)$ defines the transition. $R$ is the state and action dependent reward, and $\gamma \in (0, 1)$ is a discount factor.

For a policy $\pi$, the discounted sum of returns is denoted as a random variable $Z^{\pi}(s, a) = \sum_{t=0}^{\infty} \gamma^t R(s_t, a_t)$, where $s_0 = s$, $a_0 = a$, $s_{t+1} \sim P(\cdot|s_t, a_t)$, and $a_t \sim \pi(\cdot|s_t)$. The $Q$-value (state-action value) is defined as $Q^{\pi}(s, a) = \mathbb{E}[Z^{\pi}(s, a)]$. The optimal $Q$-value, $Q^*(s, a) = \max_{\pi} Q^{\pi}(s, a)$, is the unique fixed point of the Bellman optimality operator $\mathcal{T}$ (Bellman, 1966)

$$Q^*(s, a) = \mathcal{T}Q^*(s, a) := \mathbb{E}[R(s, a)] + \gamma \mathbb{E}_P \max_{a'} Q^*(s', a') \tag{1}$$

In most deep RL studies, $Q$ is approximated by a neural network. To update $Q$, $Q$-learning trains the network iteratively to minimize the squared temporal difference (TD) error

$$\mathcal{L}_t^2 = [r_t + \gamma \max_{a'} Q_{\phi^-}(s', a') - Q_{\phi}(s, a)]^2, \tag{2}$$

where $\phi^-$ is the target network which is updated periodically with the most recent $\phi$.

Instead of learning the scalar $Q(s, a)$, distributional RL considers the distribution over returns (the law of $Z$) to capture the aleatoric uncertainty (intrinsic stochasticity in the environment). A similar distributional Bellman operator for $Z$ can be derived as (Bellemare et al., 2017)

$$\mathcal{T}^{\pi} Z(s, a) \stackrel{D}{=} R(s, a) + \gamma Z(S', A'), \tag{3}$$

with $S' \sim P(\cdot|s, a)$, $A' \sim \pi(\cdot|S')$, and $X \stackrel{D}{=} Y$ indicates that random variables $X$ and $Y$ follow the same distribution. In theory, Bellemare et al. (2017) proved the distributional Bellman operator is a contraction in the $p$-Wasserstein metric

$$W_p(X, Y) = (\int_0^1 |F_X^{-1}(\omega) - F_Y^{-1}(\omega)|^p d\omega)^{1/p}, \tag{4}$$

where $F^{-1}$ is the quantile function (inverse cumulative distribution function). Following this theory, a series of distributional RL algorithms have been proposed based on quantile regression to estimate $F^{-1}$ at precisely chosen quantile fractions, such that the Wasserstein metric is minimized.

### 2.1 QUANTILE REGRESSION FOR DISTRIBUTIONAL RL

In QR-DQN (Dabney et al., 2018b), the random return is approximated by a uniform mixture of $N$ Diracs

$$Z_{\theta}(s, a) = \frac{1}{N} \sum_{i=1}^{N} \delta_{\theta_i(s,a)}, \tag{5}$$

with each $\theta_i$ set to a fixed quantile fraction, $\hat{\tau}_i = \frac{\tau_{i-1}+\tau_i}{2}$ for $1 \leq i \leq N$, and $\tau_i = i/N$. The quantile estimation is performed by minimizing the quantile Huber loss, with threshold $\eta$

$$\frac{1}{N} \sum_{i=1}^{N} \sum_{j=1}^{N} \rho_{\hat{\tau}_i}^{\eta}(\delta_{ij}) \tag{6}$$

on the pairwise TD error $\delta_{ij} = r + \gamma \theta_j(s', a') - \theta_i(s, a)$, where

$$\rho_{\tau}^{\eta}(\delta) = |\tau - \mathbb{I}_{\delta<0}| \mathcal{L}_{\eta}(\delta), \text{ with}$$
$$\mathcal{L}_{\eta}(\delta) = \begin{cases} \frac{1}{2}\delta^2, & |\delta| \leq \eta \\ \eta(|\delta| - \frac{1}{2}\eta), & \text{otherwise.} \end{cases} \tag{7}$$

Based on QR-DQN, Dabney et al. (2018a) proposed to sample quantile fractions from a base distribution, e.g. $\tau \in U([0,1])$ rather than fixing them. They built an implicit quantile network (IQN) to learn mappings from sampled probability embeddings to corresponding quantile values. FQF (Yang et al., 2019) further improves IQN by learning a function to propose $\tau$'s. However, the quantile values generated by neural networks may not satisfy the non-decreasing property of $F^{-1}$ (known as the quantile crossing issue). This was recently solved by NC-QR-DQN (Zhou et al., 2020), by applying a softmax, followed by a cumulative sum of the output logits of the neural network $\Omega$, and then rescaling by multiplying a non-negative factor $\alpha(s, a)$ and adding an offset $\beta(s, a)$:

$$\theta_i(s, a) = \alpha(s, a) \times \iota_{i,a} + \beta(s, a), \text{ with}$$
$$\iota_{i,a} = \sum_{j=0}^{i} \chi_{j,a}, \text{ and } \chi_{j,a} = \text{softmax}(\Omega(s))_{j,a} \tag{8}$$

One recent method NDQFN (Zhou et al., 2021) further combines the ideas of NC-QR-DQN and IQN to learn a monotonic function for $F^{-1}$ by connecting the neighboring two monotonic quantile data points with line segments. Different from NC-QR-DQN, NDQFN generates monotonic quantile values by first learning a baseline value and then adding non-negative increments.

## 2.2 OTHER DISTRIBUTIONAL METHODS

Other recent methods investigate different metrics for the distributional Bellman operator. Moment matching, generally parameterized as the maximum mean discrepancy (MMD) between two sample sets in a reproducing Hilbert kernel space (Gretton et al., 2012), is adopted by Nguyen et al. (2021) to propose moment matching DQN (MM-DQN). The MMD loss with kernel $\kappa$ is derived as:

$$d_{\kappa}^2(\{\upsilon_i\}, \{\psi_i\}) = \frac{1}{N^2} \sum_{i,j} \kappa(\upsilon_i, \upsilon_j) + \frac{1}{M^2} \sum_{i,j} \kappa(\psi_i, \psi_j) - \frac{2}{NM} \sum_{i,j} \kappa(\upsilon_i, \psi_j), \tag{9}$$

where $\{\upsilon_i\}_{i=1}^{N} \sim Z(s, a)$ and $\{\psi_i\}_{i=1}^{M} \sim R(s, a) + \gamma Z(S', A')$.

It is worth noting that the theoretical analysis by Nguyen et al. (2021) shows the distributional Bellman operator under MMD is not a contraction with commonly used Gaussian kernels or exp-prod kernels. It is a contraction only when the kernel function is shift invariant and scale sensitive.

Categorical distributional RL was also combined with policy gradient to obtain the Reactor algorithm (Gruslys et al., 2018) for discrete control and the Distributed Distributional Deep Deterministic Policy Gradient (D4PG) algorithm (Barth-Maron et al., 2018) for continuous control. Subsequently, Singh et al. (2020) replaced categorical return distributions by samples in Sample-based Distributional Policy Gradient (SDPG), yielding improved sample efficiency. The return distribution can also be represented by a generative network trained by adversarial training (in the same way as GANs) to minimize temporal differences between sampled returns (Doan et al., 2018; Freirich et al., 2019). While most distributional RL techniques compute state-action return distributions, Li & Faisal (2021) proposed the Bayesian Distributional Policy Gradient (BDPG) algorithm that computes state return distributions and uses inference to derive a curiosity bonus. In another line of work, Tessler et al. (2019) introduced the Distributional Policy Optimization (DPO) framework in which an agent's policy evolves towards a distribution over improving actions.

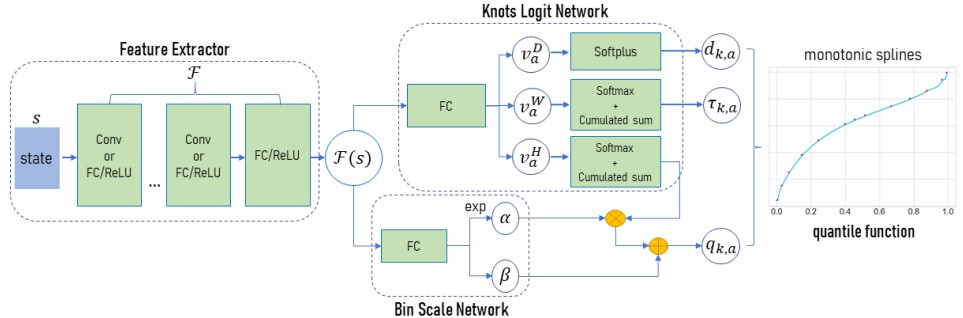

Figure 1: System Flow of SPL-DQN architecture

## 3 OUR ALGORITHM

Our method is originally motivated by NC-QR-DQN, where a special architecture is designed for the last layer of the neural network to satisfy the monotonicity of $F^{-1}$. The output represents the estimated quantile values at chosen quantile fractions. One drawback of discretization is that a precise approximation for $F^{-1}$ may need infinite fractions. But in practice, one can only use finite quantile fractions to estimate quantile values for decision making. In this work, we propose to learn a dense approximation for $F^{-1}$ using monotonic rational-quadratic splines (Gregory & Delbourgo, 1982) as a building block.

### 3.1 QUANTILE APPROXIMATION WITH MONOTONIC RATIONAL QUADRATIC SPLINES

Monotonic splines produce a monotonic interpolant to a set of monotonic data points (called knots) $\{(x_k, y_k)\}_{k=0}^K$, which has been recently used as a transformation function in normalizing flows (Durkan et al., 2019; Dolatabadi et al., 2020). Furthermore, denote $\{d_k\}_{k=0}^K$, a set of positive numbers, as the derivative of each knot. The monotonic rational-quadratic splines aim to find rational-quadratic functions with form $f_k(x) = \frac{O_k(x)}{P_k(x)}$ to fit the points and derivatives in each interval (called bin) $[x_k, x_{k+1}]$, where $O_k$ and $P_k$ are quadratic functions (with form $ax^2 + bx + c$).

Gregory & Delbourgo (1982) suggested to construct $O_k$ and $P_k$ as follows. Denote $g_k = (y_{k+1} - y_k)/(x_{k+1} - x_k)$ and $h_k(x) = (x - x_k)/(x_{k+1} - x_k)$ for $x \in [x_k, x_{k+1}]$. The expressions for the quadratic $O_k(h_k(x))$ and $P_k(h_k(x))$ for the $k^{\text{th}}$ bin is defined by (use $h_k$ for short of $h_k(x)$):

$$O_k(h_k) = g_k y_{k+1} h_k^2 + (y_k d_{k+1} + y_{k+1}) h_k (1 - h_k) + g_k y_k (1 - h_k)^2$$
$$P_k(h_k) = g_k + (d_{k+1} + d_k - 2g_k) h_k (1 - h_k) \tag{10}$$

Then, the rational-quadratic function for the $k^{\text{th}}$ bin is computed by the quotient of $O_k$ and $P_k$

$$f_k(h_k) = \frac{O_k(h_k)}{P_k(h_k)} = y_k + \frac{(y_{k+1} - y_k)[g_k h_k^2 + d_k h_k(1 - h_k)]}{g_k + (d_{k+1} + d_k - 2g_k) h_k(1 - h_k)}. \tag{11}$$

Equation 11 is proven to be monotonic and continuously differentiable, while passing through the knots and satisfying the given derivatives at the knots (Gregory & Delbourgo, 1982).

The monotonicity of the above splines fits the non-decreasing property of $F^{-1}$. Let $F_{Z(s,a)}^{-1}(\tau)$ be the quantile function for the random variable of the discounted total return $Z(s, a)$ with $\tau \in [0, 1]$. Given the number of bins $K$, the aims of the spline approximator for $F_{Z(s,a)}^{-1}$ are threefold. First, propose a partition for the domain of definition $[0, 1]$ with $\tau_0 < ... < \tau_k < ... < \tau_K$. Here $\tau_0 = 0$ and $\tau_K = 1$. Second, estimate the corresponding quantile values $q_0 < ... < q_k < ... < q_K$. Third, assess the derivatives at those points with $d_0, ..., d_k, ..., d_K$. We give a small fixed positive value for $d_0$ and $d_K$ as they are assigned with endpoints. After the generation of these three sets of statistics, the monotonic spline of each bin is given by Equation 11 (by replacing $x_k$ by $\tau_k$ and $y_k$ by $q_k$).

### 3.2 MODEL IMPLEMENTATION

We now show how to learn the monotonic splines for quantile functions in distributional RL by neural networks, and we name the technique *spline DQN* (SPL-DQN). As shown in Figure 1, the

SPL-DQN consists of three major components, including a *Feature Extractor* which extracts latent features from a state, a *Knots Logit Network* which, for each action, generates the logits of the widths and heights for $K$ bins, and derivatives for $K - 1$ inner knots, and a *Bin Scale Network* which recovers the heights in $[0, 1]$ to the original quantile range. Here we describe the model for a discrete action space of size $|\mathcal{A}|$. To use monotonic splines in continuous control, the model can be modified by taking state-action pairs as input and only producing knots for that state-action pair.

The *Feature Extractor* $\mathcal{F}$ is usually made up of multiple convolutional layers with subsequent fully-connected layers for image-like inputs or stacked fully-connected layers for non-image inputs. It produces the feature embedding $\mathcal{F}(s) \in \mathbb{R}^d$ of state $s$. Then the *Knots Logit Network* $\mathcal{W}$ maps $\mathcal{F}(s)$ to unconstrained logits $v$ with dimension $|\mathcal{A}| \times (3K - 1)$ using a fully-connected layer. The vector $v_a$ for each action $a$ is partitioned as $v_a = [v_a^W, v_a^H, v_a^D]$, where $v_a^W$ and $v_a^H$ have length $K$, and $v_a^D$ has length $K - 1$. Instead of directly learning $\tau_{k,a}$ and $q_{k,a}$ associated with each monotonic knot, we propose to learn the normalized width and height of each bin. Here, vectors $v_a^W$ and $v_a^H$ are each passed through a softmax function and are interpreted as the normalized widths and heights. Vector $v_a^D$ is regarded as the derivatives, and is passed through a softplus function to satisfy monotonicity.

With the width and height of each bin, $\tau_{k,a}$ and $q_{k,a}$ of each knot can be easily calculated by a cumulative sum. Since the values of $v_a^W$ and $v_a^H$ fall into $[0, 1]$, each $\tau_{k,a}$ ($k > 0$) is computed by

$$\tau_{k,a} = \sum_{i=1}^{k} v_{i,a}^W, \ k = 1, ..., K; \ a = 1, ..., |\mathcal{A}| \tag{12}$$

without rescaling as the domain of a quantile function is $[0, 1]$ ($\tau_{0,a} = 0$ by definition). To compute each $q_{k,a}$, another transformation is required to rescale $v_a^H$ to a range corresponding to the true quantile values. Inspired by NC-QR-DQN, we introduce the *Bin Scale Network* to generate two adaptive scale factors $\alpha$ and $\beta$ by applying a fully connected layer $\mathcal{C}: \mathbb{R}^d \to \mathbb{R}^{|\mathcal{A}| \times 2}$ to the state embedding $\mathcal{F}(s)$. We compute the exponential of $\alpha$ to ensure the total bin height is positive. Then $q_{0,a} = \beta_a$ and for $k > 0$, $q_{k,a}$ is computed by

$$q_{k,a} = \exp(\alpha_a) \times \sum_{i=1}^{k} v_{i,a}^H + \beta_a, \ k = 1, ..., K; \ a = 1, ..., |\mathcal{A}| \tag{13}$$

### 3.3 Approximate Wasserstein Metric Minimization

When using continuous approximations of the quantile functions, there are several choices to compute the integral of the Wasserstein metric between two quantile functions. We can try to calculate the integral directly, but this is not straightforward for rational-quadratic functions since the integral rarely has a closed form. Alternatively, we can calculate the Riemann integral, but this leads to a loss function analogous to the L1-norm, which may cause instability in training. Thus, in this work, we perform quantile regression (Koenker & Hallock, 2001) in a projected space to approximately minimize the Wasserstein metric.

Let $\tilde{\tau} = (\tilde{\tau}_0, ..., \tilde{\tau}_N)$ be a fixed sequence of non-decreasing quantile fractions (note that the set $\tilde{\tau}$ is different from the $x$-values of knots in Section 3.1 to partition the $[0, 1]$ domain, which are learned by the neural network. In our experiments, we let $\tilde{\tau}$ be uniformly fixed), we project the monotonic spline quantile function $f$ to a quantile distribution space $\mathcal{Z}_Q$ by computing

$$Z_q(s, a) = \sum_{i=1}^{N} (\tilde{\tau}_i - \tilde{\tau}_{i-1}) \delta_{\hat{q}_i(s,a)}, \tag{14}$$

where each $\hat{q}_i(s, a)$ is the corresponding quantile value at the quantile fraction $\hat{\tau}_i = \frac{\tilde{\tau}_{i-1} + \tilde{\tau}_i}{2}$ given by $f(\hat{\tau}_i)$ with $1 \leq i \leq N$. To compute $f(\hat{\tau}_i)$, we first search which bin $\hat{\tau}_i$ lies in. Then the value is returned by the corresponding spline function given $h_k(\hat{\tau}_i)$ as input. In this case, the optimal value distribution $Z$ is achieved by minimizing the 1-Wasserstein metric with $Z_q$

$$W_1(Z(s, a), Z_q(s, a)) = \sum_{i=1}^{N} \int_{\tilde{\tau}_{i-1}}^{\tilde{\tau}_i} |F_{Z(s,a)}^{-1}(\omega) - \hat{q}_i(s, a)| d\omega, \tag{15}$$

which is equivalent to finding a projection operator $\Pi_{W_1}$ such that

$$\Pi_{W_1} Z := \arg \min_{Z_q \in \mathcal{Z}_Q} (Z, Z_q). \tag{16}$$

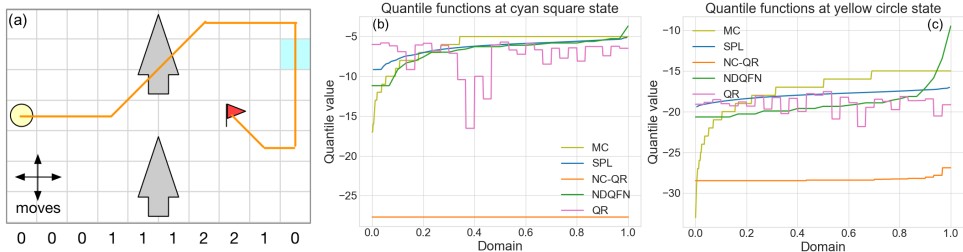

Figure 2: (a) Windy Gridworld, with wind strength shown along bottom row. (b) & (c) The quantile functions for value distribution of the cyan square state and yellow circle state by MC, SPL-DQN (SPL), NC-QR-DQN (NC-QR), NDQFN, and QR-DQN (QR).

Furthermore, Dabney et al. (2018b) shows that the unique minimizer of this operator is given by

$$F_{Z(s,a)}^{-1}(\hat{\tau}_i) = \hat{q}_i(s,a), \ \hat{\tau}_i = \frac{\tilde{\tau}_{i-1} + \tilde{\tau}_i}{2} \tag{17}$$

**Proposition 1.** [Proposition 2 in Dabney et al. (2018b)] *Let $\Pi_{W_1}$ be the quantile projector defined above. When applied to value distributions, it gives a projection for each state-value distribution. For any two value distributions $Z_1, Z_2 \in \mathcal{Z}$ for an MDP with countable state and action spaces,*

$$\overline{d}_\infty(\Pi_{W_1}\mathcal{T}^\pi Z_1, \Pi_{W_1}\mathcal{T}^\pi Z_2) \leq \gamma \overline{d}_\infty(Z_1, Z_2), \tag{18}$$

*where $\overline{d}_p(Z_1, Z_2) = \sup_{s,a} W_p(Z_1(s,a), Z_2(s,a))$ and $\mathcal{Z}$ is the space of action-value distributions with finite moments.*

Proposition 1 suggests that after projecting $f$ to $Z_q$, the operator $\Pi_{W_1}\mathcal{T}^\pi$ is a $\gamma$-contraction under the measure $\overline{d}_\infty$ and the repetition of this operator converges to a fixed point in space $\mathcal{Z}_Q$.

Based on Proposition 1, the ultimate goal is to estimate quantile values in Equation 17 for $F_{Z(s,a)}^{-1}$ using quantile regression in each training batch. In our implementation, we uniformly fix $\tilde{\tau} = (\tilde{\tau}_0, ..., \tilde{\tau}_N)$ to be consistent with QR-DQN and NC-QR-DQN, which leads to the same quantile Huber loss as shown in Equations 6 and 7. However, the advantage of our method over QR-DQN and NC-QR-DQN is that we can freely enrich the density of $\tilde{\tau}$ to get a better estimation of the quantile function without increasing the size of the model architecture, while QR-DQN and NC-QR-DQN must enlarge the output dimension of their models to get more quantile estimates. Since we can freely query quantile values at any quantile fraction, quantile fraction embedding as done in IQN and FQF is no longer necessary in our method.

**Remark**: Although one recent method, NDQFN, also learns continuous monotonic quantile functions, our method is different from NDQFN in three aspects. First, the $x$-values of those monotonic knots, i.e., $\tau_0, ..., \tau_K$, are uniformly fixed in NDQFN, while they are trainable in our method. Second, by also learning the derivatives at each knot, we get a smooth interpolant over the entire domain, while NDQFN connects those knots with line segments, which has limited approximation ability. Third, to get the increments of $y$-values of those knots, i.e., $q_0, ..., q_K$, NDQFN learns a function taking the quantile fraction embeddings, i.e., the embeddings of corresponding $\tau$s, as input, while we do not calculate increments but use a scale network as discussed in Section 3.2.

To demonstrate the monotonicity and approximation strength of our method in stochastic environments, we plot the quantile functions learned by SPL-DQN, NC-QR-DQN, NDQFN, and QR-DQN in a variant of the classic Windy Gridworld domain (Sutton & Barto, 2018). In Figure 2a, the agent starts at the yellow circle state and makes standard moves in a gridworld to reach the red flag. A reward of $-1$ is earned at each step. Some columns are affected by some wind blowing from bottom to top. The orange line shows the optimal trajectory without stochasticity. We set each state transition to have probability 0.1 of moving in a random direction without any wind effect, otherwise the transition is affected by the wind, which pushes the agent northward. All methods here use the same training settings and similar network architectures as discussed in Appendix A.1. We compute the ground truth value distribution for an optimal policy (learned by policy iteration) at each state by performing one thousand Monte-Carlo (MC) rollouts and recording the observed returns as an empirical distribution. Then we transform the empirical distribution to the quantile function as the baseline. Here we show case the learned quantile functions at cyan square state and yellow circle state (start state) as shown in Figures 2b and 2c.

All these four methods eventually learn the optimal policy, however their quantile approximations are quite different. Without constraints, quantile functions given by QR-DQN clearly violate the monotonic property, which is known as the quantile crossing issue (Zhou et al., 2020). Although NC-QR-DQN applies monotonic constraints, the estimated quantile range is biased towards smaller values according to the quantile functions given by MC, and we observe that the quantile functions learned by NC-QR-DQN are straight lines for some states, e.g. cyan square state, which means that it fails to learn the value distribution in those states, and in turn this leads to a biased estimation for the start state. The reason for this biased estimation is that in NC-QR-DQN, when rescaling the quantile range in Equation 8, a ReLU function is imposed to the coefficient $\alpha$ to ensure it is non-negative. However, this often sets $\alpha$ to zero and the quantile distribution will only depend on the shift parameter $\beta$ (which leads to a straight line). In this case, the value distribution cannot be precisely captured. For NDQFN, its quantile approximation at the goal nearing state (cyan square state) is close to SPL-DQN , but it overestimates the quantile range at the start state. We also observe the overestimation issue of NDQFN at another state in the middle of the orange line trajectory as shown in Figure 6 in the appendix. Though still exhibiting estimation errors, the quantile functions learned by SPL-DQN are often the closest to the ground truth.

## 4 EXPERIMENTS

While most previous distributional RL algorithms were evaluated with Atari games from the Arcade Learning Environment (ALE), it was noted that the ALE is deterministic (Bellemare et al., 2017) and therefore questionable as a benchmark to evaluate distributional algorithms that are designed to capture environment stochasticity when there is none. However, we note that sticky actions can be used in Atari games to introduce stochasticity in policies (Machado et al., 2018) and this regime was used to evaluate IQN and FQF (Yang et al., 2019). When the environment is deterministic, value distributions still arise due to stochastic policies, stochastic approximations and random parameter initialization, but the resulting value distributions tend to be simple and close to deterministic. It is also well-known that deterministic environments possess optimal

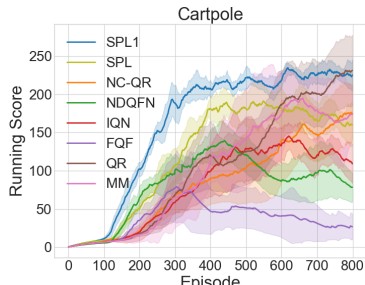

Figure 3: Performance comparison in stochastic Cartpole. Each curve is averaged over 5 seeds.

policies that are open-loop and therefore ignore observations (Machado et al., 2018; Koul et al., 2019). In practice, it is often desirable to train controllers with simulators in which noise is injected to increase the robustness of the learned policies in case of discrepancies between the simulator and the real world. Hence, in this work, we modify several robotics environments by adding stochasticity, including one discrete environment from OpenAI Gym (Brockman et al., 2016) and nine continuous environments from PyBulletGym (Ellenberger, 2018–2019). We compare our method with QR-DQN, IQN, FQF, NC-QR-DQN, MM-DQN, and NDQFN. For MM-DQN, we used the unrectified Kernel $\kappa_\alpha(x, y) = -||x - y||^\alpha$ with $\alpha = 1$ (parameter taken from Nguyen et al. (2021)) instead of the Gaussian kernel recommended by the authors when they tested on Atari games since the unrectified kernel gave better results in the robotics benchmarks used in this paper. A summary of how different QR-based methods compute the QR loss is provided in Appendix B. For a fair comparison, we made sure the same *Feature Extractor* architecture was used in different models. To simplify acronyms, we omit -DQN when referring to a method in what follows.

### 4.1 DISCRETE CONTROL IN CARTPOLE

We begin our experimental results in a stochastic environment with a discrete action space modified from Cartpole (Florian, 2007). The system is controlled by a force of $+1$ or $-1$ applied to the cart. A reward of $+1$ is returned if the pole remains upright. We set each state transition to have probability 0.05 of moving to a neighboring state to make the environment stochastic. The QR-based methods use $N = 8$ quantiles to compute the QR loss. MM-DQN uses $N = M = 8$ samples. More training details are provided in Appendix A.2.

As the episode rewards may vary significantly due to stochasticity, to better reflect the training process, we define the running score as a soft update of episode rewards:

$$running\_score = 0.99 \times running\_score + 0.01 \times episode\_rewards \qquad (19)$$

Figure 3 shows the running score curves for stochastic Cartpole. In general, SPL learns much faster (faster empirical convergence) than its counterparts. As discussed before, SPL can freely increase the number of quantiles when performing quantile regression without enlarging the output dimension of the model. We further increase the number of quantiles to 24 to compute the QR loss while keeping the number of bins unchanged ($K = 8$ and $N = 24$), yielding the curve labeled 'SPL1' in Figure 3. This curve shows that approximately minimizing the Wasserstein metric with more quantiles leads to better quantile approximations and increases the learning speed and performance of SPL. As NDQFN also learns continuous quantile functions, we do the same experiment ($K = 8$ and $N = 24$) for NDQFN, whose training curve is labeled by 'NDQFN1' in Figure 7 in the appendix. Although its training performance improves, SPL with $N = 24$ is still better.

## 4.2 CONTINUOUS CONTROL IN PYBULLETGYM

PyBulletGym provides RoboSchool[1], which is a free port of MuJoCo[2]. The state of these environments contains joint information of a robot and an action is a multi-dimensional continuous vector. We take nine environments from RoBoSchool and make them stochastic by introducing Gaussian noise $\mathcal{N}(\mu, \sigma)$ to both the location and velocity of each part of the robot, with $\mu = 0$ and $\sigma$ varying in different environments. We choose a reasonable $\sigma$ for each environment such that robots won't exhibit unrealistic motion. That is, for noise sensitive environments, such as Walker2D and Humanoid, we use a smaller $\sigma$, and for relatively easy tasks, like InvertedPendulumSwingup, we choose a bigger one. The noise setting for different environments is shown in Table 3 in the appendix.

To evaluate on continuous control tasks, we combine distributional RL with DDPG (Lillicrap et al., 2016) by modifying the critic, as done by Zhang & Yao (2019). Instead of learning $Q$, the critic learns the distribution $Z$ directly. To handle continuous actions, the critic takes state-action pairs as input. As an exception, for the Humanoid environment, we combine distributional RL with SAC (Haarnoja et al., 2018) due to the fact that DDPG is not as good as SAC for this environment. To update the actor in DDPG and SAC, the expectation of $Q$ values is computed as the expectation of quantile samples given by the distributional critic. We refer to the original papers for hyperparameter settings, which are discussed in Appendix A.3. We also include raw DDPG and SAC as baselines.

Figure 4 shows the running score curves given by Equation 19 for these stochastic environments. Generally, the training performance varies among different approaches in different environments, however, in most cases, the quantile regression based methods who learn monotonic quantile representations are better than those whose quantile representations have no monotonicity guarantee, which clarifies that the quantile crossing issue can distort policy learning as pointed out by Zhou et al. (2020). Especially, for SPL, apart from Reacher and InvertedPendulumSwingup, it always converges faster and performs better during training. For InvertedPendulumSwingup, SPL performs comparably to NC-QR. Although NDQFN also learns continuous monotonic quantile functions, its performance is even worse than NC-QR in most cases, because NDQFN queries linear functions for quantile samples when computing QR loss, but the approximation ability of piecewise linear function is very limited. For methods with no monotonic quantile guarantee, we notice that although IQN is the best in Reacher, it performs worse in InvertedPendulum and Humanoid during training.

To further demonstrate the ability of our method to handle uncertainty of the environment, we slightly increase the noise in HalfCheetah to $\mathcal{N}(0, 0.008)$ (labeled by HalfCheetah1) and $\mathcal{N}(0, 0.01)$ (labeled by HalfCheetah2). The training curves in these two environments are shown in Figure 5. On average, QR and MM for DDPG behave poorly in these three HalfCheetah variants. The enhanced randomness of environments degrades the training performance of SPL, but SPL is generally faster and better than NC-QR and IQN, thanks to more precise quantile approximations.

During training, Ornstein-Uhlenheck noise $\mathcal{OU}(\mu', \sigma')$ (Uhlenbeck & Ornstein, 1930) is utilized when selecting actions to induce exploration in DDPG. At the evaluation stage, the methods are executed with only exploitation (without action noise). We test the best models we get after training for each method, and the testing score across different environments are shown in Table 6 in Appendix A.3. Apart from Reacher, SPL outperforms its counterparts in all other domains. For the first two environments, although the training performances vary significantly among different methods, the testing scores of their best models are close to each other. For most remaining environments, the

---

[1]https://openai.com/blog/roboschool/
[2]http://www.mujoco.org

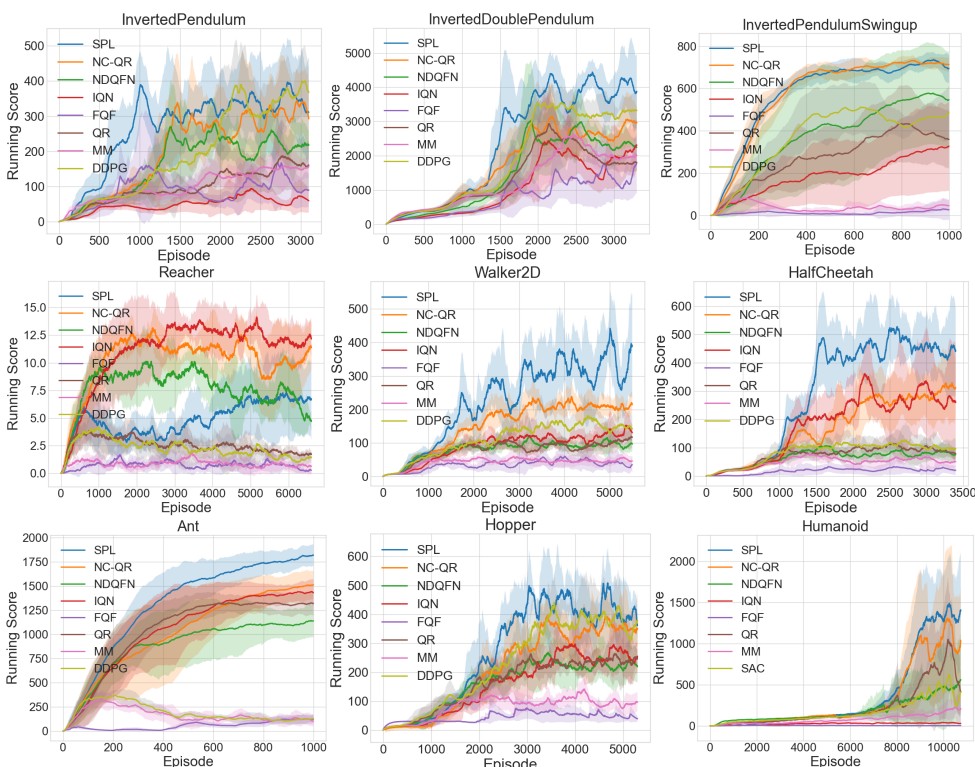

Figure 4: Performance comparison in stochastic RoboSchool. Each curve is averaged by 7 seeds. The first eight environments are solved with DDPG. The last one is assigned to SAC.

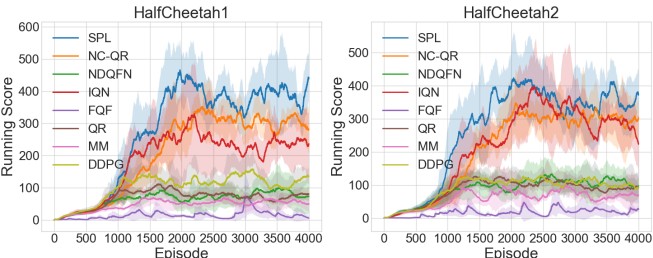

Figure 5: Performance comparison in two stochastic HalfCheetahs with enhanced randomness.

testing scores of SPL and NC-QR are significantly better than other methods. Appendix C provides additional results for SPL and NDQFN when trained with uniformly spaced quantile fractions and random quantile fractions sampled from $U([0, 1])$.

**Limitations:** Our technique assumes full state observability. This is a limitation that could be removed in future work by extending it to partially observable domains. Similar to previous work in distributional RL, there is a computational cost associated with representing the value distribution. Hence our technique does not scale as well as non-distributional RL techniques. While we demonstrated our spline approximation in combination with DQN, DDPG and SAC, our distributional RL technique is limited to RL techniques that include a critic.

## 5  CONCLUSION

Based on previous works in distributional RL, we propose a more general and precise approximation for quantile functions using monotonic rational-quadratic splines when minimizing the Wasserstein metric. With a monotonic continuous representation of the quantile function, the quantile value at every quantile level is accessible during training, yielding greater accuracy. One direction for future work is to investigate other monotonic function approximators for quantile estimation.

## ACKNOWLEDGMENTS

We acknowledge the funding from the Canada CIFAR AI Chairs program, the Natural Sciences and Engineering Research Council of Canada (NSERC) and support from SportLogiq. Resources used in this work were provided, in part, by the Province of Ontario, the Government of Canada through CIFAR, and companies sponsoring the Vector Institute `https://vectorinstitute.ai/partners/`.

## ETHICS STATEMENT

We confirm that our work has no ethics issue.

## REPRODUCIBILITY STATEMENT

To ensure reproducibility, we have discussed the detailed experiment settings in Appendix A. The code for the main experiments is released in the supplementary material.

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

## A   EXPERIMENT DETAILS

### A.1   WINDY GRIDWORLD

**Training hyperparameters** We train SPL-DQN with NC-QR-DQN, NDQFN, and QR-DQN in the stochastic Windy Gridworld. The common parameter settings are summarized in Table 1. The model inputs are the coordinates of the state. The $\epsilon$-greedy parameter decreases by a half every two thousand episodes. To make the results comparable, we use the same Feature Extractor for the methods. For SPL-DQN and NDQFN, the number of bins is $K = 30$.

Table 1: Common hyperparameters for SPL-DQN, NC-QR-DQN, NDQFN, and QR-DQN.

| Hyperparameter | Value |
| --- | --- |
| Optimizer | Adam |
| Learning rate | 0.001 |
| Batch size | 50 |
| Discount factor ($\gamma$) | 1 |
| Initial $\epsilon$-greedy | 0.3 |
| Minimal $\epsilon$-greedy | 0.01 |
| Training episodes | 30000 |
| Sampling quantiles number | 30 |
| Feature Extractor hidden size | [20, 40, 80] |

**Additional results** The learned quantile functions for the green square state are shown in Figure 6. NDQFN overestimates the quantile range.

To further demonstrate the approximation ability of SPL compared with NDQFN, we train SPL and NDQFN with 10 random seeds in windy gridworld, and check the quantile functions learnt by these two methods for the states on the orange line trajectory (apart from the goal state). There are 15 states on the trajectory times 10 seeds, which yield 150 quantile functions. For each state, we can use the shortest path to determine an upper bound on the return for that state. Whenever a quantile function goes above that upper bound, we have a clear overestimation. For NDQFN, $36.7\%$ $(55/150)$ of the quantile functions yield an overestimation. In contrast, for SPL, only $4\%$ $(6/150)$ of the quantile functions yield an overestimation.

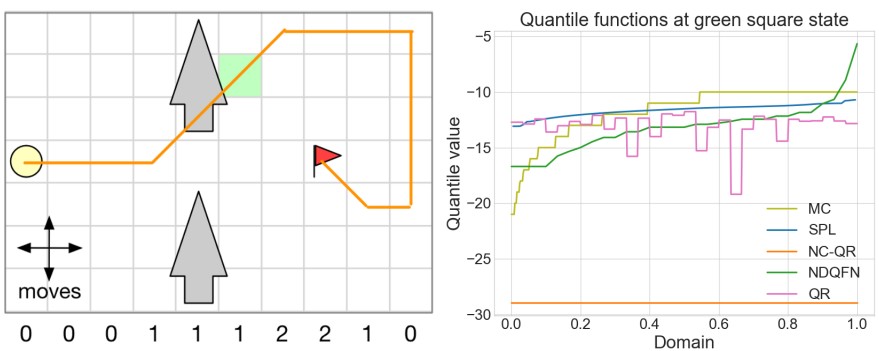

Figure 6: The learned quantile functions at the green square state

### A.2   CARTPOLE

**Training hyperparameters** We train SPL-DQN with QR-DQN, IQN, FQF, NC-QR-DQN, MM-DQN, and NDQFN in the stochastic Cartpole. The common parameter settings are summarized in Table 2. The model input is a vector of length 4, which contains the cart position, the cart velocity, the pole angle, and the pole angular velocity. The $\epsilon$-greedy parameter decreases by $0.00005$ every time step. To make the results comparable, we use the same Feature Extractor for the methods.

Specifically, for SPL-DQN and NDQFN, the number of bins is $K = 8$. For FQF, it contains a quantile fraction proposal network, whose learning rate is $2.5e^{-9}$, and the optimizer is RMSProp (Yang et al., 2019).

Table 2: Common hyperparameters across SPL-DQN, QR-DQN, IQN, FQF, NC-QR-DQN, MM-DQN, and NDQFN

| Hyperparameter | Value |
|---|---|
| Optimizer | Adam |
| Learning rate | 0.001 |
| Batch size | 32 |
| Discount factor ($\gamma$) | 0.99 |
| Initial $\epsilon$-greedy | 0.3 |
| Minimal $\epsilon$-greedy | 0.1 |
| Training episodes | 800 |
| Sampling quantiles number (QR based methods) | 8 |
| Samples number (MM-DQN) | 8 |
| Feature Extractor hidden size | [128, 128] |

**Additional results** As NDQFN also learns continuous quantile functions, the number of quantile samples can be enlarged when computing the QR loss. Here we train NDQFN with bin number $K = 8$ and quantile number $N = 24$. The training curve is labeled by 'NDQFN1' as shown in Figure 7. The curves labeled by 'SPL1', 'SPL', and 'NDFQN' are taken from Figure 3.

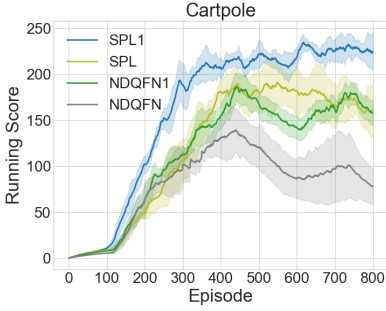

Figure 7: Performance comparison in stochastic Cartpole

## A.3 PYBULLETGYM

**Noise setting for different environments** We introduce different noise levels in PyBulletGym's environments while ensuring that the robots won't exhibit unrealistic motion. The noise settings are shown in Table 3.

Table 3: Noise settings for different environments in PyBulletGym

| Environments | Noise |
|---|---|
| InvertedPendulum | $\mathcal{N}(0, 0.02)$ |
| InvertedDoublePendulum | $\mathcal{N}(0, 0.01)$ |
| InvertedPendulumSwingup | $\mathcal{N}(0, 0.05)$ |
| Reacher | $\mathcal{N}(0, 0.01)$ |
| Walker2D | $\mathcal{N}(0, 0.005)$ |
| HalfCheetah | $\mathcal{N}(0, 0.005)$ |
| HalfCheetah1 | $\mathcal{N}(0, 0.008)$ |
| HalfCheetah2 | $\mathcal{N}(0, 0.01)$ |
| Ant | $\mathcal{N}(0, 0.01)$ |
| Hopper | $\mathcal{N}(0, 0.003)$ |
| Humanoid | $\mathcal{N}(0, 0.003)$ |

**Training hyperparameters** Hyperparameters for DDPG and DDPG based models are summarized in Table 4. The critic also uses an $L_2$ weight decay of $10^{-2}$. The soft target update coefficient is 0.001. Ornstein-Uhlenheck noise ($\mathcal{OU}(\mu', \sigma')$) (Uhlenbeck & Ornstein, 1930) is combined with actions for exploration in DDPG, where we use $\mu' = 0$ and $\sigma' = 0.1$.

Hyperparameters for SAC and SAC based models are summarized in Table 5. The soft target update coefficient is 0.005.

For the critic implemented by SPL-DQN and NDQFN, the number of bins is $K = 32$. For the critic implemented by FQF, the learning rate for quantile fraction network is $2.5e^{-9}$, and the corresponding optimizer is RMSProp.

Table 4: Hyperparameters for DDPG and DDPG based methods

| Hyperparameter | Value |
| --- | --- |
| Optimizer | Adam |
| Actor learning rate | $10^{-4}$ |
| Critic learning rate | $10^{-3}$ |
| Batch size | 64 |
| Discount factor ($\gamma$) | 0.99 |
| Training frames | one million |
| Sampling quantiles number | 32 |
| Actor hidden size | [400, 300] |
| Critic's Feature Extractor hidden size | [400, 300] |

Table 5: Hyperparameters for SAC and SAC based methods

| Hyperparameter | Value |
| --- | --- |
| Optimizer | Adam |
| Actor learning rate | $3 \times 10^{-3}$ |
| Critic learning rate | $3 \times 10^{-3}$ |
| Entropy learning rate | $3 \times 10^{-3}$ |
| Batch size | 64 |
| Discount factor ($\gamma$) | 0.99 |
| Training frames | three million |
| Sampling quantiles number | 32 |
| Actor hidden size | [256, 256] |
| Critic's Feature Extractor hidden size | [256, 256] |

**Additional results** We test the best models given by different methods with four random seeds. The results are shown in Table 6. We test all DDPG based agents without Ornstein-Uhlenheck noise for 0.125 million frames, and SAC based agents for 2.5 thousand episodes. We treat DDPG and SAC scores as baselines and scale other methods' scores by them, i.e.

$$method\_scaled\_test\_score = \frac{method\_raw\_test\_score}{DDPG/SAC\_raw\_test\_score} \tag{20}$$

## B  COMPUTING QR LOSS

We summarize how different QR-based distributional RL methods sample quantile values when computing the QR loss. Since the QR loss is computed in a TD manner, we will need $N$ current quantile samples ($\{q_i^1\}, i = 1, ..., N$) and $N'$ target quantile samples ($\{q_i^2\}, i = 1, ..., N'$) corresponding to two quantile fraction sets ($\{\tau_i^1\}, i = 1, ..., N$) and ($\{\tau_i^2\}, i = 1, ..., N'$). Without loss of generality, we consider $N = N'$. Here we discuss the case with discrete actions, and denote the action space by $|\mathcal{A}|$.

For discrete quantile approximations, including QR-DQN, NC-QR-DQN, IQN, and FQF, in order to get $N$ quantile samples for each action, the output dimension of the model is $|\mathcal{A}| \times N$ for an input

Table 6: Scaled testing scores across different stochastic environments. Scores are averaged over 4 seeds.

| Environments | MM | QR | FQF | IQN | NDQFN | NC-QR | SPL |
|---|---|---|---|---|---|---|---|
| InvertedPendulum | 0.911 | 0.940 | 0.953 | 0.970 | 0.992 | 0.969 | **0.999** |
| InvertedDoublePendulum | 0.814 | 0.978 | 0.975 | 0.967 | 0.990 | 0.993 | **1.019** |
| InvertedPendulumSwingup | 0.461 | 0.945 | 0.223 | 0.944 | 1.091 | 1.145 | **1.179** |
| Reacher | -1.501 | 0.412 | -10.546 | **4.269** | 3.416 | 4.241 | 2.972 |
| Walker2D | 0.503 | 0.661 | 0.585 | 1.375 | 0.776 | 1.732 | **3.142** |
| HalfCheetah | 0.731 | 1.084 | 0.809 | 2.122 | 1.039 | 2.932 | **3.004** |
| HalfCheetah1 | 0.763 | 0.897 | 0.859 | 1.764 | 1.156 | 2.158 | **2.633** |
| HalfCheetah2 | 0.834 | 0.855 | 0.773 | 1.741 | 1.231 | 1.812 | **2.21** |
| Ant | 0.871 | 2.283 | 0.345 | 2.403 | 2.388 | 3.045 | **3.321** |
| Hopper | 0.689 | 0.868 | 0.671 | 0.960 | 0.893 | 1.405 | **1.609** |
| Humanoid | 1.077 | 1.409 | 0.035 | 0.044 | 1.108 | 1.558 | **1.640** |

state. For QR-DQN and NC-QR-DQN, $\{\tau_i^1\}$ and $\{\tau_i^2\}$ are assumed to be uniformly spaced. For IQN, $\{\tau_i^1\}$ and $\{\tau_i^2\}$ are independently drawn from a uniform distribution $U([0,1])$. For FQF, $\{\tau_i^1\}$ and $\{\tau_i^2\}$ are proposed by a quantile fraction network.

For methods that learn a continuous approximation of the quantile function, including SPL-DQN and NDQFN, the output (for an input state) consists of knots with shape $|\mathcal{A}| \times (K+1)$ when the domain is divided into $K$ bins. For SPL-DQN, it leanrs the $x$, $y$ values, and derivatives of those knots. A smooth continuous function with closed form is obtained in each bin. When sampling quantile values to compute the QR loss, SPL-DQN uniformly fixes $\{\tau_i^1\}$ and $\{\tau_i^2\}$, and $\{q_i^1\}$ and $\{q_i^2\}$ are obtained by querying the closed form with $\{\tau_i^1\}$ and $\{\tau_i^2\}$ as inputs. For NDQFN, it only learns the $y$-values of those knots, and the $x$-values of the knots are uniformly spaced. The continuous function is constructed by connecting neighboring knots with linear functions. When sampling quantile values to compute the QR loss, NDQFN draws $\{\tau_i^1\}$ and $\{\tau_i^2\}$ from a uniform distribution $U([0,1])$ independently, and $\{q_i^1\}$ and $\{q_i^2\}$ are obtained by querying the linear functions in each bin.

## C    COMPARISON OF SPL AND NDQFN WITH TWO TRAINING REGIMES

Figure 8 shows additional results that compare SPL and NDQFN when trained with uniformly spaced points versus points sampled uniformly at random. In this figure, SPL corresponds to training with uniformly spaced quantile fractions as described in the main paper. NDQFN corresponds to training with quantile fractions sampled from $U([0,1])$ as described in (Zhou et al., 2021). To iron out this difference in the training, we also report SPL-rnd where training is done with quantile fractions sampled from $U([0,1])$ and NDQFN-uni where training is done with uniformly spaced quantile fractions. We observe that SPL outperforms NDQFN in both training regimes.

## D    DDPG AND SAC WITH DISTRIBUTIONAL CRITIC

We summarize the algorithm when using distributional critic for DDPG and SAC in this section. For FQF critic, it has to update the quantile fraction proposal network separately, so it is individually described. Similarly, for MM critic, it uses moment matching instead of quantile regression, and it is individually described as well. For SAC based methods, readers can refer to the original paper for how to compute gradient for policy and state value.

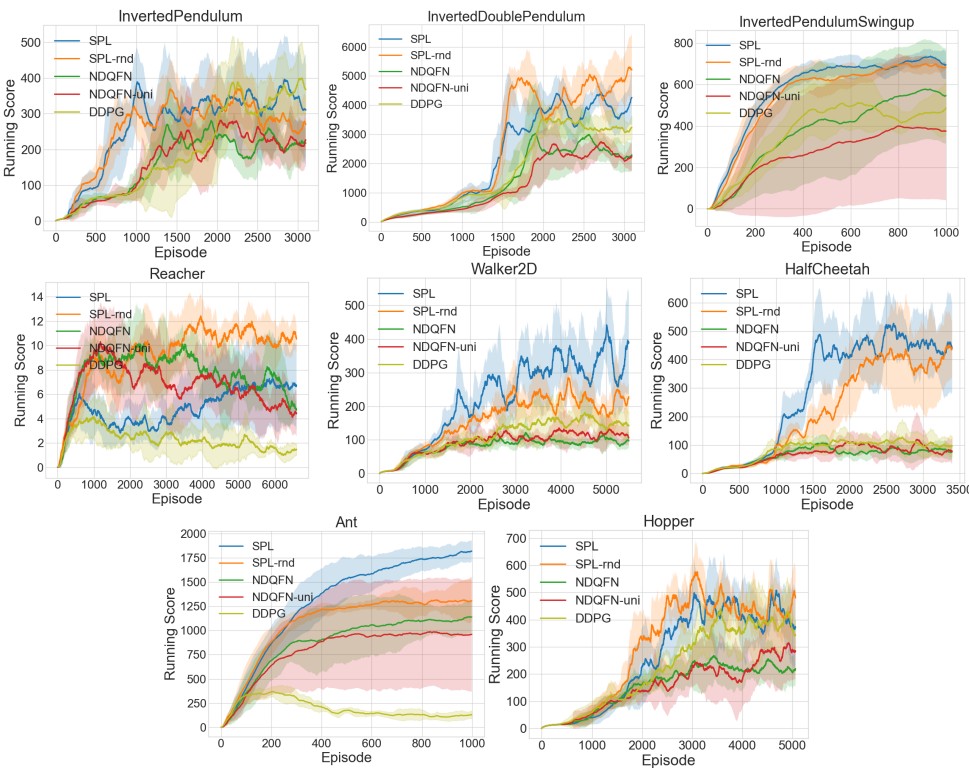

Figure 8: Performance comparison of SPL and NDQFN when trained with uniformly spaced quantile fractions or random quantile fractions sampled from $U([0, 1])$ in eight environments with DDPG as the baseline

---

**Algorithm 1** DDPG with QR-based distributional critic (apart from FQF)

**Require:** Initialize critic network $Z(s, a|\psi)$ with weights $\psi$; Initialize actor network $\mu(s|\theta)$ with weights $\theta$; Initialize target network $Z'$ and $\mu'$ with weights $\psi' \leftarrow \psi$, $\theta' \leftarrow \theta$; Initialize reply buffer $\mathcal{D}$

1: **for** each episode **do**
2:     Initialize random process $\mathcal{OU}$ for action exploration
3:     **for** each time step t **do**
4:         $a_t \sim \mu(s_t|\theta) + \mathcal{OU}_t$
5:         $s_{t+1} \sim p(s_{t+1}|s_t, a_t)$
6:         $\mathcal{D} \leftarrow \mathcal{D} \cup \{(s_t, a_t, r(s_t, a_t), s_{t+1})\}$
7:         Sample minibatch of $N$ transitions $(s_i, a_i, r_i, s_{i+1})$ from $\mathcal{D}$
8:         Choose current quantile fractions $\{\tau\}$ according to critic's strategy
9:         Compute corresponding current quantiles $\{q_i\} \leftarrow Z(s_i, a_i|\psi)$
10:       Choose target quantile fractions $\{\bar{\tau}\}$ according to critic's strategy
11:       Compute corresponding target quantiles $\{\bar{q}_{i+1}\} \leftarrow r_i + \gamma Z'(s_{i+1}, \mu'(s_{i+1}|\theta')|\psi')$
12:       Update critic by minimizing QR loss
13:       Compute expectation of quantiles $Q_{s,a} \leftarrow \mathbb{E}[Z(s, a|\psi)]$
14:       Update actor by
15:         $\nabla_\theta J \approx \frac{1}{N} \sum_i \nabla_a Q_{s,a}|_{s=s_i, a=\mu(s_i)} \nabla_\theta \mu(s|\mu)|_{s_i}$
16:       Update target networks
17:         $\psi' \leftarrow \sigma\psi + (1 - \sigma)\psi'$
18:         $\theta' \leftarrow \sigma\theta + (1 - \sigma)\theta'$
19:     **end for**
20: **end for**

---

**Algorithm 2** DDPG with QR-based distributional critic (FQF)

---

**Require:** Initialize critic value network $Z(s, a|\psi)$ with weights $\psi$; Initialize actor network $\mu(s|\theta)$ with weights $\theta$; Initialize target network $Z'$ and $\mu'$ with weights $\psi' \leftarrow \psi$, $\theta' \leftarrow \theta$; Initialize reply buffer $\mathcal{D}$

1: Initialize critic fraction proposal network $P(s, a|\omega)$ with weights $\omega$ and its target $P'(s, a|\omega')$ with weights $\omega'$
2: **for** each episode **do**
3:     Initialize random process $\mathcal{OU}$ for action exploration
4:     **for** each time step t **do**
5:         $a_t \sim \mu(s_t|\theta) + \mathcal{OU}_t$
6:         $s_{t+1} \sim p(s_{t+1}|s_t, a_t)$
7:         $\mathcal{D} \leftarrow \mathcal{D} \cup \{(s_t, a_t, r(s_t, a_t), s_{t+1})\}$
8:         Sample minibatch of $N$ transitions $(s_i, a_i, r_i, s_{i+1})$ from $\mathcal{D}$
9:         Compute current quantile fractions $\{\tau\} \leftarrow P(s_i, a_i|\omega)$
10:        Compute corresponding current quantiles $\{q_i\} \leftarrow Z(s_i, a_i|\psi)$
11:        Choose target quantile fractions $\{\bar{\tau}\} \leftarrow P(s_{i+1}, \mu'(s_{i+1}|\theta')|\omega)$
12:        Compute corresponding target quantiles $\{\bar{q}_{i+1}\} \leftarrow r_i + \gamma Z'(s_{i+1}, \mu'(s_{i+1}|\theta')|\psi')$
13:        Update critic fraction proposal network by
14:           $\frac{\partial \omega}{\partial \tau_i} = 2Z[\tau_i] - Z[\hat{\tau}_i] - Z[\hat{\tau}_{i-1}]$, $\hat{\tau} = \frac{\tau_i + \tau_{i+1}}{2}$
15:        Update critic value network by minimizing QR loss
16:        Compute expectation of quantiles $Q_{s,a} \leftarrow \mathbb{E}[Z(s, a|\psi)]$
17:        Update actor by
18:           $\nabla_\theta J \approx \frac{1}{N} \sum_i \nabla_a Q_{s,a}|_{s=s_i, a=\mu(s_i)} \nabla_\theta \mu(s|\mu)|_{s_i}$
19:        Update target networks
20:           $\psi' \leftarrow \sigma\psi + (1 - \sigma)\psi'$
21:           $\theta' \leftarrow \sigma\theta + (1 - \sigma)\theta'$
22:           $\omega' \leftarrow \sigma\omega + (1 - \sigma)\omega'$
23:     **end for**
24: **end for**

---

**Algorithm 3** DDPG with MM distributional critic

---

**Require:** Initialize critic network $Z(s, a|\psi)$ with weights $\psi$; Initialize actor network $\mu(s|\theta)$ with weights $\theta$; Initialize target network $Z'$ and $\mu'$ with weights $\psi' \leftarrow \psi$, $\theta' \leftarrow \theta$; Initialize reply buffer $\mathcal{D}$

1: **for** each episode **do**
2:     Initialize random process $\mathcal{OU}$ for action exploration
3:     **for** each time step t **do**
4:         $a_t \sim \mu(s_t|\theta) + \mathcal{OU}_t$
5:         $s_{t+1} \sim p(s_{t+1}|s_t, a_t)$
6:         $\mathcal{D} \leftarrow \mathcal{D} \cup \{(s_t, a_t, r(s_t, a_t), s_{t+1})\}$
7:         Sample minibatch of $N$ transitions $(s_i, a_i, r_i, s_{i+1})$ from $\mathcal{D}$
8:         Compute current Q samples $\{q_i\} \leftarrow Z(s_i, a_i|\psi)$
9:         Compute target Q samples $\{\bar{q}_{i+1}\} \leftarrow r_i + \gamma Z'(s_{i+1}, \mu'(s_{i+1}|\theta')|\psi')$
10:        Update critic by minimizing MMD loss
11:        Compute expectation of Q values $Q_{s,a} \leftarrow \mathbb{E}[Z(s, a|\psi)]$
12:        Update actor by
13:           $\nabla_\theta J \approx \frac{1}{N} \sum_i \nabla_a Q_{s,a}|_{s=s_i, a=\mu(s_i)} \nabla_\theta \mu(s|\mu)|_{s_i}$
14:        Update target networks
15:           $\psi' \leftarrow \sigma\psi + (1 - \sigma)\psi'$
16:           $\theta' \leftarrow \sigma\theta + (1 - \sigma)\theta'$
17:     **end for**
18: **end for**

---

---

**Algorithm 4** SAC with QR-based distributional critic (apart from FQF)

---

**Require:** The learning rates $\lambda_\pi$, $\lambda_Z$, and $\lambda_V$ for functions $\pi_\theta$, $Z_w$, and $V_\psi$; Initialize parameters $\theta$, $w$, $\psi$, $\bar{\psi}$; Initialize reply buffer $\mathcal{D}$

1: **for** each iteration **do**
2:     **for** each time step t **do**
3:         $a_t \sim \pi_\theta(s_t)$
4:         $s_{t+1} \sim p(s_{t+1}|s_t, a_t)$
5:         $\mathcal{D} \leftarrow \mathcal{D} \cup \{(s_t, a_t, r(s_t, a_t), s_{t+1})\}$
6:     **end for**
7:     **for** each gradient update step **do**
8:         Choose current and target quantile fractions according to critic's strategy
9:         Compute current and target quantile values (for computing $J_Z$)
10:        $\psi \leftarrow \psi - \lambda_V \nabla_\psi J_V(\psi)$
11:       $w_i \leftarrow w_i - \lambda_Z \nabla_{w_i} J_Z(w_i)$ for $i \in \{1, 2\}$ ($J_Z$ is QR loss)
12:       $\theta \leftarrow \theta - \lambda_\pi \nabla_\theta J_\pi(\theta)$
13:       $\bar{\psi} \leftarrow \sigma\psi + (1 - \sigma)\bar{\psi}$
14:     **end for**
15: **end for**

---

**Algorithm 5** SAC with QR-based distributional critic (FQF)

---

**Require:** The learning rates $\lambda_\pi$, $\lambda_Z$, and $\lambda_V$ for functions $\pi_\theta$, $Z_w$, and $V_\psi$; Initialize parameters $\theta$, $w$, $\psi$, $\bar{\psi}$; Initialize reply buffer $\mathcal{D}$; The learning rate $\lambda_P$ for fraction proposal network $P_\phi$; Initialize parameter $\phi$

1: **for** each iteration **do**
2:     **for** each time step t **do**
3:         $a_t \sim \pi_\theta(s_t)$
4:         $s_{t+1} \sim p(s_{t+1}|s_t, a_t)$
5:         $\mathcal{D} \leftarrow \mathcal{D} \cup \{(s_t, a_t, r(s_t, a_t), s_{t+1})\}$
6:     **end for**
7:     **for** each gradient update step **do**
8:         Compute current and target quantile fractions using $P_\phi$
9:         Compute current and target quantile values (for computing $J_Z$)
10:        $\psi \leftarrow \psi - \lambda_V \nabla_\psi J_V(\psi)$
11:       $\phi \leftarrow \phi - \lambda_\phi \nabla_\phi J_P(\phi)$ ($J_P$ is quantile fraction loss)
12:       $w_i \leftarrow w_i - \lambda_Z \nabla_{w_i} J_Z(w_i)$ for $i \in \{1, 2\}$ ($J_Z$ is QR loss)
13:       $\theta \leftarrow \theta - \lambda_\pi \nabla_\theta J_\pi(\theta)$
14:       $\bar{\psi} \leftarrow \sigma\psi + (1 - \sigma)\bar{\psi}$
15:     **end for**
16: **end for**

---

**Algorithm 6** SAC with QR-based distributional critic (MM)

---

**Require:** The learning rates $\lambda_\pi$, $\lambda_Z$, and $\lambda_V$ for functions $\pi_\theta$, $Z_w$, and $V_\psi$; Initialize parameters $\theta$, $w$, $\psi$, $\bar{\psi}$; Initialize reply buffer $\mathcal{D}$

1: **for** each iteration **do**
2:     **for** each time step t **do**
3:         $a_t \sim \pi_\theta(s_t)$
4:         $s_{t+1} \sim p(s_{t+1}|s_t, a_t)$
5:         $\mathcal{D} \leftarrow \mathcal{D} \cup \{(s_t, a_t, r(s_t, a_t), s_{t+1})\}$
6:     **end for**
7:     **for** each gradient update step **do**
8:         Compute current and target Q value samples (for computing $J_Z$)
9:        $\psi \leftarrow \psi - \lambda_V \nabla_\psi J_V(\psi)$
10:       $w_i \leftarrow w_i - \lambda_Z \nabla_{w_i} J_Z(w_i)$ for $i \in \{1, 2\}$ ($J_Z$ is MMD loss)
11:       $\theta \leftarrow \theta - \lambda_\pi \nabla_\theta J_\pi(\theta)$
12:       $\bar{\psi} \leftarrow \sigma\psi + (1 - \sigma)\bar{\psi}$
13:     **end for**
14: **end for**

---

