# OpenReview forum: "Distributional Reinforcement Learning with Monotonic Splines"
_ICLR.cc/2022/Conference — ICLR 2022 Poster_

### Official Review · Reviewer_zyuH · 2021-10-28

**Correctness:** 3
**Technical Novelty And Significance:** 2
**Empirical Novelty And Significance:** 2
**Recommendation:** 6
**Confidence:** 5

**Main Review:**

Strengths:
- the proposed parameterization is continuously differentiable, in contrast to previous piecewise linear approaches
- paper is clearly written and well positioned with respect to prior work
- experimental results show significant improvements over previous approaches in many environments with added noise to assess robustness

Weaknesses:
- novelty is low with respect to NDQFN, which uses piecewise linear interpolation instead
- it is a bit disappointing that it ends up considering a fixed set of quantile levels for the loss computation, which makes the whole approach equivalent to a fancy parameterization of a fixed level quantile network
- it lacks some kind of analysis of the origin of the improvement over NDQFN.


### Detailed comments:
It is claimed that this approach "offers greater accuracy in terms of quantile approximation" and is "a more general and precise approximation for quantile functions" .
These claims seem to be based on the visual assessment of Figure 2. I think they should be more strongly supported with some approximation result comparing splines to piecewise linear functions, with an equivalent number of degrees of freedom.
However, it is not clear, how a better capability to approximate an arbitrary function is useful since only the quantile values at the fixed quantile levels matter when optimizing the quantile loss.
Regarding the improvement over NDQFN: How much comes from differentiability ? How much comes from the representation itself (splines vs piecewise linear)? How much comes from the different choice of quantile levels  (fixed in SPL-DQN, and uniformly sampled in NDQFN)?  Also, a greater number of trainable parameters in SPL could also explain the increased performance.
Also, why isn't  SPL always superior (cf. Reacher experiment)?


### Minor issues/comments:
Eq (4) : large parentheses should be used

"During training, ${\cal OU}(\mu';\sigma')$ noise..." : this notation should be introduced

The combination with DDPG/SAC should be described (at least in the appendix).


**Summary Of The Paper:**

This paper proposes a new neural network design to represent quantile functions for distributional reinforcement learning, based on smooth rational-quadratic splines. This representation has the advantage of being continuously differentiable.
The loss is computed by evaluating the quantile loss on a set of uniformly spaced quantile levels.
The benefits are shown empirically on standard continuous control environments, modified with noise to make them stochastic, by using the proposed distributional representation as critic in DDPG/SAC algorithms.

**Summary Of The Review:**

The method is a natural "upgrade" of NDQFN and thus its novelty is relatively low.
The experimental results on continuous control environments with noise show better results in most environments.
However, it lacks a real analysis of the causes of this improvement.

For these reasons, I tend to vote for rejection in this current form.


**================== After rebuttal ==================**

Although I find the spline approach and the experimental results interesting,
I still feel that the paper lacks theory to explain the improvement and, also, that novelty is relatively low.
My concern regarding the fixed quantile levels used for training has been addressed with the uniform sampling and the new experiments.

Therefore, I increase my score to 6.

---

> ### Author Response · Authors · 2021-11-17
> **Reply to reviewer zyuH**
>
> We thank the reviewer for the detailed review as well as the suggestions for improvement.
>
> **Regarding the weakness**
>
> 1. *"it is a bit disappointing that it ends up considering a fixed set of quantile levels for the loss computation, which makes the whole approach equivalent to a fancy parameterization of a fixed level quantile network"*
>
> Choosing uniformly fixed quantile level is just a general way to sample quantile values, which is already adopted by QR-DQN and NC-QR-DQN. We also adopt this sampling technique, since computing the expectation of Q values is more straightforward. In addition, we can always enrich the quantile level density with little cost to get more quantile samples (by querying the spline function), so either uniformly sampling or randomly sampling is not a very big deal here.
>
> For "makes the whole approach equivalent to a fancy parameterization of a fixed level quantile network", we do not think it is correct. Since each query is computed by Eq. 11 with the statistics of the corresponding bin, the gradient of the training loss will back propagate to those bin statistics. Thus, what our model learns are really the $x$, $y$, and derivative of each knots, not the fixed level quantiles.
>
> 2. *"it lacks some kind of analysis of the origin of the improvement over NDQFN"*
>
> The analysis is originally provided on page 6 of the paper, the paragraph beginning with **Remark**.
>
> In summary, first, NDQFN fixes the $x$-values of the knots, while they are learnt in SPL, which increases the flexibility. Second, quantile functions learnt by NDQFN are not smooth while those obtained by SPL are.
>
>
> **Regarding the detailed comments**
>
> 1.*"It is claimed that this approach "offers greater accuracy in terms of quantile approximation" and is "a more general and precise approximation for quantile functions" . These claims seem to be based on the visual assessment of Figure 2. I think they should be more strongly supported with some approximation result comparing splines to piecewise linear functions, with an equivalent number of degrees of freedom"*
>
> Generally, regardless of the smoothness, the piecewise linear functions are within the representation space of the rational quadratic functions, i.e., by setting the numerator to be linear and the denominator to be constant. In addition, we also take into account derivatives, such that the learned function is smooth and more like an inverse cdf function. Thus the rational quadratic parameterization is a more general representation than the piecewise linear parameterization.
>
> 2. *"Also, a greater number of trainable parameters in SPL could also explain the increased performance."*
>
> In our implementation of NDQFN and SPL, the number of parameters of these two methods are in the same magnitude.
>
> **Regarding the minor issues**
>
> 1. *"During training, OU noise..." : this notation should be introduced*
>
> We add the reference in the main paper.  See the new version of the paper.
>
> 2 *"The combination with DDPG/SAC should be described (at least in the appendix)."*
>
> We add the algorithm description in the appendix. See the new version of the paper.

---

> > ### Comment · Reviewer_zyuH · 2021-11-22
> > **Comment on fixed quantile level projection**
> >
> > I agree that the model learns $x$,$y$ and the derivative of each knot, but it fits them in order to estimate the fixed level quantiles. Therefore $x$,$y$ and the derivatives can be seen as parameters for the fixed level quantiles.
> > If, instead of considering a fixed set of quantile levels, you randomly sample them for each batch as you suggest in your response, then the situation would be different since the full quantile function would be used throughout the training instead of just its projection (14).
> > I think that this possibility should be discussed in the paper.

---

> > > ### Author Response · Authors · 2021-11-23
> > > **Regarding fixed quantile level projection**
> > >
> > > 1. *"I agree that the model learns $x$, $y$ and the derivative of each knot, but it fits them in order to estimate the fixed level quantiles. Therefore $x$, $y$, and the derivatives can be seen as parameters for the fixed level quantiles"*
> > >
> > > Thank you for raising this concern.
> > >
> > > It is useful to make an analogy here.  Let's consider a general regression problem where a neural network is used to approximate some unknown underlying function.  In practice, we define a loss function that minimizes the error at some data points.  We can definitely think of the parameters of the neural networks as simply being parameters used to fit the target values of those data points.  However, we do not think of the neural network as only estimating those data points.  They happen to be the data points used for the loss function, but the neural network is really approximating the underlying function everywhere.  In fact, at test time, the neural network is used to make predictions at other points than those used during training.  Furthermore, we can't associate specific parameters with each training data point since each training data point may depend on all the parameters.
> > >
> > > Similarly, SPL learns a continuous underlying function in the form of a rational quadratic spline.  To train this rational quadratic spline, we minimize the QR-loss at uniformly spaced data points.  We can think of those uniformly spaced data points as the training points used by the loss function.  Those data points are different from the knots and their x, y and derivatives that we learn.  We agree that the target quantile values of the training points can be formulated in terms of the x, y and derivatives of the knots.  This is just like the target values of the training data points that can be expressed as functions of the parameters of a neural network in general regression.  However, the representation learned by SPL is not restricted to the training data points in the loss function since we can output quantile values for any input.  In the paragraph above the remark on page 6, we explain how we can "freely enrich the density of $\tilde{\tau}$ to get a better estimation of the quantile
> > > function".  This is essentially saying that we can make predictions at other data points than the training points when approximating the integral by discretizing.  Note that this is not the case for previous methods (NC-QR-DQN, QR-DQN) since they can only make predictions at the training points.
> > >
> > > To summarize, methods that learn fixed quantile values can only make predictions at those training points.  In contrast, SPL learns a rational quadratic spline by minimizing a loss at some training points that are chosen to be the same as previous methods for a fair comparison, but SPL can make predictions anywhere (not just the training points).  To illustrate this point, we include in the appendix of the paper additional results where SPL uses sampled points instead of uniformly spaced points as requested by the reviewer.
> > >
> > > 2. *"If, instead of considering a fixed set of quantile levels, you randomly sample them for each batch as you suggest in your response, then the situation would be different since the full quantile function would be used throughout the training instead of just its projection (14). I think that this possibility should be discussed in the paper."*
> > >
> > > We agree that when we sample points at random during training then the full quantile function is fitted in the limit of infinite training.  Following your suggestion, we included experiments that compare what happens when we train with uniformly spaced quantile levels versus random quantile levels for SPL and NDQFN.

---

### Official Review · Reviewer_Gqyh · 2021-10-30

**Correctness:** 4
**Technical Novelty And Significance:** 3
**Empirical Novelty And Significance:** 3
**Recommendation:** 6
**Confidence:** 5

**Main Review:**

**Novelty**: The idea of using monotonic rational-quadratic splines to interpolate an inverse cdf function is novel and interesting. I enjoyed reading this paper.

**Correctness**: The methods are sound and derivations are correct.

**Technicity**: Though the idea is novel, the technicity in the paper is quite low as the paper applied existing techniques to the quantile-crossing problem and combined several existing elements. The most significant technicity in the paper is perhaps the neural architecture to model the monotonic rational-quadratic splines, though this modelling is quite natural.

**Clarity**: The paper is well written and organized. One comment for improving the clarity is that in defining $f_k$ in Eq. (11), it is more clearer to say that $h \in [x_k, x_{k+1}]$, and replacing $h$ in the RHS of Eq. (11) by $(h - x_k) / (x_{k+1} - x_k)$. The reason is that in the paragraph after Eq. (14), the paper talked about computing $f(\hat{\tau_i})$ and searching for the bin of $\hat{\tau_i}$. In the original form, $\hat{\tau_i} = h(x)$ which is not possible to search for a bin with this representation.

**Experimental significance**: The paper demonstrated that their method, when used in actor-critic framework for continuous control, better approximates the inversed CDF and better performance than the other distributional RL methods in most experiments. What I am concerned about is that why didn't the authors test their algorithm on Atari as most distributional RL methods did, so that the comparison is more standard? Of course, the Atari testbed has its own problem as the paper also discussed but it is a nice testbed for distributional RL because only valued-based RL methods are compared, this performance is more reflective of whether the value estimation is good. In the actor-critic framework, the final performance also depends on the actor; thus if SPL+actor > IQN + actor, it is not clear if SPL alone is still better than IQN alone in value-based RL setting.

**Summary Of The Paper:**

This paper proposes to use monotonic rational-quadratic splines to interpolate an inverse cdf function for distributional RL. They proposed a neural architecture to implement such interpolation and achieved a simplified yet effective solution to the quantile crossing problem in distributional RL.

**Summary Of The Review:**

The paper makes a novel contribution in terms of using monotonic rational-quadratic splines to interpolate an inverse cdf function for distributional RL. The empirical result is very promising. I suggest weak acceptance.

---

> ### Author Response · Authors · 2021-11-16
> **Reply to reviewer Gqyh**
>
> We thank the reviewer for detailed review as well as the suggestions for improvement.
>
> **Regarding the clarity**
>
> 1. *"One comment for improving the clarity is that in defining $f_k$ in Eq. (11),  it is more clearer to say that $h\in[x_k, x_{k+1}]$"*
>
> Thanks very much for your suggestion. The rational behind our original definition of $h$ was to make Eq. (11) simpler and clearer. Taking into account your suggestion, we propose to keep the current definition of $h$, while changing the notation to $h_{k}(x)$ to improve clarity.
>
> **Regarding the experiments**
>
> 1. *"What I am concerned about is that why didn't the authors test their algorithm on Atari as most distributional RL methods did, so that the comparison is more standard? Of course, the Atari testbed has its own problem as the paper also discussed but it is a nice testbed for distributional RL because only valued-based RL methods are compared, this performance is more reflective of whether the value estimation is good. In the actor-critic framework, the final performance also depends on the actor; thus if SPL+actor > IQN+actor, it is not clear if SPL alone is still better than IQN alone in value-based RL setting."*
>
> We understand that most previous distributional RL techniques were evaluated on Atari, but as pointed in Section 4 and acknowledged by the reviewer, Atari is deterministic and therefore questionable.  As a community, we should not stick with deterministic domains just because previous algorithms were evaluated on deterministic domains.  The right thing to do is to test on stochastic domains since distributional RL is meant for stochastic domains. Note that it was more work for us to test on domains that previous distributional RL techniques had not been tested on since we had to reimplement previous algorithms, but we did it since it is the right thing to do.  The community should be encouraged to test on stochastic domains as we did going forward.
>
> For actor-critic methods, we use the same parameters and training settings for all the actors to ensure a fair comparison. We also compared plain SPL to other value based techniques in stochastic windy gridworld and cartpole (Figures 2 and 3).

---

### Official Review · Reviewer_2kiV · 2021-11-02

**Correctness:** 4
**Technical Novelty And Significance:** 4
**Empirical Novelty And Significance:** 4
**Recommendation:** 8
**Confidence:** 3

**Main Review:**

The paper is well written and clearly presented. Related works are surveyed in details. Although some existing works have already solved the quantile crossing problem, the paper carefully discusses the difference of such methods and the proposed algorithm, thereby justifying its novelty. In empirical evaluation, some of the standard experimental environment have been modified with randomness injected, and the paper provides clear motivation for such modification (more suitable for evaluating distributional RL algorithms). The extensive experiment results show that the proposed algorithm outperforms the existing ones in most of the test cases and therefore justify its effectiveness beyond novelty.

**Summary Of The Paper:**

The paper studies the problem of distributional reinforcement learning. It augments the traditional quantile-based algorithms with monotonic rational-quadratic splines. Such augmentation provides a natural solution to the quantile crossing issue, which exists for many other quantile-based algorithms. Extensive experiment results are performed to verify the effectiveness of the proposed method.

**Summary Of The Review:**

The paper proposes a novel algorithm with strong empirical results.

---

> ### Author Response · Authors · 2021-11-16
> **Reply to reviewer 2kiV**
>
> We thank the reviewer for the positive feedback.

---

### Official Review · Reviewer_uP3K · 2021-11-02

**Correctness:** 3
**Technical Novelty And Significance:** 2
**Empirical Novelty And Significance:** 2
**Recommendation:** 5
**Confidence:** 4

**Main Review:**

Originality: SPL-DQN provides an alternative way of parameterizing the quantile function with monotonicity in distributional RL. Specifically, using an architecture (feature extractor + logit network + bin scale network) similar to NC-QR-DQN, SPL-DQN substantiates the non-crossing quantile logit network by using the rational-quadratic splines.

While the idea of using rational-quadratic splines is interesting, I do find that the proposed method needs to be better motivated:
- The benefits of using monotonic rational-quadratic splines as quantile approximators need to be better justified. In Section 3.3, empirical comparison of the four QR-DQN-based methods is provided, and it is mentioned that NC-QR-DQN could suffer from underestimation and degenerate quantile functions (e.g., quantile function being a straight line) in some states, and NDQFN could suffer from overestimation. While I appreciate the above observations, it remains unclear to me whether these observations are indeed fundamental systematic issues caused by the algorithm design. More empirical evidence or qualitative explanation (e.g., which design leads to the observed estimation bias) is required to strengthen these arguments.
- On the other hand, I think a more detailed comparison between IQN and SPL-DQN is needed, given that IQN could already represent a general class of quantile functions without the quantile crossing problem. Is there any qualitative benefit of using SPL-DQN compared to IQN (besides comparing them vis-a-vis in the experiments)?


Significance: The experimental results show that combining SPL with DDPG improves over the existing benchmark distributional RL methods in multiple tasks of Roboschool. Having said that, in many tasks (e.g., Walker2D, Halfcheetah, Hopper, and Reacher) the improvement of SPL over other monotonic QR-DQN counterparts appears somewhat marginal compared to the benchmarking results (e.g., https://github.com/araffin/rl-baselines-zoo/blob/master/benchmark.md). One possibility is that such performance differences could partially result from the additional stochasticity added to the environment. To clarify this, it could be helpful to add other stronger baselines like TD3 and SAC as a reference.

On the other hand, to strengthen the empirical significance of SPL-DQN per se (as SPL as a critic can perform quite differently compared to SPL-DQN), it would be really helpful to provide an empirical comparison in more complex environments with discrete actions (e.g., Atari or MinAtar) in spite of the issue of stochasticity in the first paragraph of Section 4. This could better validate if the original SPL-DQN is indeed strong compared to other distributional RL methods.

Clarity: The overall writing and organization of the paper are good. The proposed method is in most places well-explained. There are only a few cases where additional details are provided in the appendix.


**Summary Of The Paper:**

This paper proposes SPL-DQN, a quantile-based distributional RL method that uses monotonic rational-quadratic splines to approximate the quantile function of the cumulative return (starting from each state-action pair), and the features of using this technique are: (i) The approximated quantile function is ensured to be monotonic; (ii) The monotonic rational-quadratic splines are continuously differentiable and can provide a more flexible class of smooth approximators for the quantile function (compared to piecewise linear or step functions).
Moreover, the feature (i) resolves the quantile crossing problem of several existing distributional RL methods (e.g. QR-DQN), under which the quantiles output from the neural network may not be monotonic. By conducting experiments with a toy example, this paper also empirically discovers a few potential issues with the relevant prior works that enforce monotonicity (e.g., NC-QR-DQN and NDQFN). Experimental results in both tasks with discrete (Cartpole) and continuous action spaces (Roboschool) are provided to demonstrate the performance of SPL-DQN.

**Summary Of The Review:**

This paper proposes a new variant of QR-DQN with ensured monotonicity in the quantile function. Given that the quantile crossing issue is not new and the design of SPL-DQN closely follows NC-QR-DQN, the overall technical novelty is somewhat limited. Despite that SPL appears to outperform other variants of QR methods in some control tasks, the amount of improvement is not very significant compared to the benchmarking results. Therefore, I lean towards rejection for now but am willing to change my score if the authors address the concerns.

---

> ### Author Response · Authors · 2021-11-16
> **Reply to reviewer uP3K**
>
> We thank the reviewer for the detailed review as well as the suggestions for improvement.
>
> **Regarding the quantile approximation of NC-QR-DQN**:
>
> 1. *"it is mentioned that NC-QR-DQN could suffer from underestimation and degenerate quantile functions (e.g., quantile function being a straight line) in some states"*
>
> The analysis for NC-QR-DQN is discussed at the top of page 7. NC-QR-DQN tries to rescale values in the [0,1] range to the true quantile range by multiplying a parameter $\alpha$ and then adding a shift parameter $\beta$, as shown in Equation 8. Here, $\alpha$ and $\beta$ are learned by two neural networks. However, to ensure monotonicity, NC-QR-DQN applies a ReLU function to $\alpha$, which may set $\alpha$ to 0, and the quantile values will all become $\beta$. In this case, the quantile function is a straight line, as shown in Figure 2(b). This approximation error will propagate to other states during training (Figure 2(c)).
>
> **Regarding comparison between IQN and SPL-DQN**:
>
> 1. *"On the other hand, I think a more detailed comparison between IQN and SPL-DQN is needed, given that IQN could already represent a general class of quantile functions without the quantile crossing problem. Is there any qualitative benefit of using SPL-DQN compared to IQN"*
>
> As far as we know, there is no monotonicity guarantee in IQN. IQN takes quantile fraction embeddings as input and output corresponding quantile values. It still has the quantile crossing problem as pointed out by Fan Zhou et al., 2021.
>
> The benefit of our method over IQN is that, first, monotonicity of quantile function is guaranteed in our method, second, we learn an approximation of the quantile function by approximating target quantile values and their derivatives whereas IQN only tries to approximate target quantile values.
>
> [Fan Zhou et al., 2021]  Non-decreasing quantile function network with efficient exploration for distributional reinforcement learning. IJCAI, 2021.
>
> **Regarding the experiments**
>
> 1. *"the improvement of SPL over other monotonic QR-DQN counterparts appears somewhat marginal compared to the benchmarking results. One possibility is that such performance differences could partially result from the additional stochasticity added to the environment. To clarify this, it could be helpful to add other stronger baselines like TD3 and SAC as a reference."*
>
> Since we add randomness to the environment, the tasks become harder than the original deterministic versions, thus the benchmarking scores in the absence of any noise are not suitable references.
>
> We have included SAC as a baseline in Humanoid environment (the last graph of Figure 4). In this case, the original critic of SAC is replaced by other distributional versions. Note that our approach can improve the result of any technique that uses a critic by making it distributional as shown in Figure 4 with DDPG and SAC.
>
> 2. *"it would be really helpful to provide an empirical comparison in more complex environments with discrete actions (e.g., Atari or MinAtar) in spite of the issue of stochasticity in the first paragraph of Section 4"*
>
> As a community, we need to test distributional RL algorithms on stochastic environments.  As explained in Section 4 and acknowledged by the reviewer, Atari is deterministic and therefore questionable.  Thank you for pointing out MinAtar, which includes some stochasticity but is a simplified version of a few Atari games.  Hence, it is not clear that MinAtar is necessarily more complex or a better benchmark than the stochastic robotics environments that we tested with. The bottom line is that we tested the algorithms on stochastic domains and we encourage the community to do the same since it is the right thing to do to evaluate distributional RL algorithms.

---

> > ### Author Response · Authors · 2021-11-19
> > **Reply to reviewer uP3K - Part 2**
> >
> > **Regarding the quantile approximation of NDQFN**
> >
> > 1. *"NDQFN could suffer from overestimation" in windy gridworld*
> >
> > We train SPL and NDQFN with 10 random seeds in windy gridworld, and check the quantile functions learnt by these two methods for the states on the orange line trajectory (apart from the goal state). There are 15 states on the trajectory times 10 seeds, which yield 150 quantile functions. For each state, we can use the shortest path to determine an upper bound on the return for that state.  Whenever a qunatile function goes above that upper bound, we have a clear overestimation.  For NDQFN, 36.7 percent (55/150) of the quantile functions yield an overestimation.  In contrast, for SPL, only 4 percent (6/150) of the quantile functions yield an overestimation.

---

### Public Comment · ~Kashif_Rasul1 · 2022-04-29
**Related works**

I believe the following two papers are relevant related works that come from the prob. forecasting approach:

1. Probabilistic Forecasting with Spline Quantile Function RNNs http://proceedings.mlr.press/v89/gasthaus19a.html
2. Learning Quantile Functions without Quantile Crossing for Distribution-free Time Series Forecasting https://arxiv.org/pdf/2111.06581.pdf
3. Learning Quantile Functions for Temporal Point Processes with Recurrent Neural Splines https://souhaib-bentaieb.com/papers/2022_RQS.pdf

---

### Decision · Program_Chairs · 2022-01-20

**Decision:**

Accept (Poster)

**Comment:**

The paper proposes monotonic splines as an improvement on current approaches to parametrising quantiles in distributional RL. The idea is an obvious, natural improvement on what exists, and yields improved experimental results.